

# Selection of a marker gene to construct a reference library for wetland plants, and the application of metabarcoding to analyze the diet of wintering herbivorous waterbirds

Yuzhan Yang[1], Aibin Zhan[2], Lei Cao[2], Fanjuan Meng[2] and Wenbin Xu[3]

[1] School of Life Sciences, University of Science and Technology of China, Hefei, Anhui, China
[2] Research Center for Eco-Environmental Sciences, Chinese Academy of Sciences, Beijing, China
[3] Anhui Shengjin Lake National Nature Reserve Administration, Chizhou, Anhui, China

Corresponding author
Lei Cao, caolei@ustc.edu.cn

## ABSTRACT

Food availability and diet selection are important factors influencing the abundance and distribution of wild waterbirds. In order to better understand changes in waterbird population, it is essential to figure out what they feed on. However, analyzing their diet could be difficult and inefficient using traditional methods such as microhistologic observation. Here, we addressed this gap of knowledge by investigating the diet of greater white-fronted goose *Anser albifrons* and bean goose *Anser fabalis*, which are obligate herbivores wintering in China, mostly in the Middle and Lower Yangtze River floodplain. First, we selected a suitable and high-resolution marker gene for wetland plants that these geese would consume during the wintering period. Eight candidate genes were included: *rbc*L, *rpo*C1, *rpo*B, *mat*K, *trn*H-*psb*A, *trn*L (UAA), *atp*F-*atp*H, and *psb*K-*psb*I. The selection was performed via analysis of representative sequences from NCBI and comparison of amplification efficiency and resolution power of plant samples collected from the wintering area. The *trn*L gene was chosen at last with c/h primers, and a local plant reference library was constructed with this gene. Then, utilizing DNA metabarcoding, we discovered 15 food items in total from the feces of these birds. Of the 15 unique dietary sequences, 10 could be identified at specie level. As for greater white-fronted goose, 73% of sequences belonged to *Poaceae* spp., and 26% belonged to *Carex* spp. In contrast, almost all sequences of bean goose belonged to *Carex* spp. (99%). Using the same samples, microhistology provided consistent food composition with metabarcoding results for greater white-fronted goose, while 13% of *Poaceae* was recovered for bean goose. In addition, two other taxa were discovered only through microhistologic analysis. Although most of the identified taxa matched relatively well between the two methods, DNA metabarcoding gave taxonomically more detailed information. Discrepancies were likely due to biased PCR amplification in metabarcoding, low discriminating power of current marker genes for monocots, and biases in microhistologic analysis. The diet differences between two geese species might indicate deeper ecological significance beyond the scope of this study. We concluded that DNA metabarcoding provides new perspectives for studies of herbivorous waterbird diets and inter-specific interactions, as well as new

possibilities to investigate interactions between herbivores and plants. In addition, microhistologic analysis should be used together with metabarcoding methods to integrate this information.

## INTRODUCTION

Wetlands are one of the most important ecosystems in nature, and they harbor a variety of ecosystem services such as protection against floods, water purification, climate regulation and recreational opportunities (*Brander, Florax & Vermaat, 2006*). Waterbirds are typically wetland-dependent animals upon which they could get abundant food and suitable habitats (*Ma et al., 2010*). Waterbird abundance and distribution could reflect the status of wetland structure and functions, making them important bio-indicators for wetland health (*Fox et al., 2011*). Among all factors affecting waterbird community dynamics, food availability is frequently considered to play one of the most important roles (*Wang et al., 2013*). However, recently suitable food resources have tended to decrease or even disappear due to deterioration and loss of natural wetlands (*Fox et al., 2011*). As a result, waterbirds are forced to discard previous habitats and sometimes even feed in agricultural lands (*Zhang et al., 2011*). In addition, migratory waterbirds may aid the dispersal of aquatic plants or invertebrates by carrying and transporting them between water bodies at various spatial scales (*Reynolds, Miranda & Cumming, 2015*). Consequently, long-time monitoring and systematic studies of waterbird diets are essential to understand population dynamics of waterbirds, as well as to establish effective management programs for them (*Wang et al., 2012*).

Traditional methods for waterbird diet analysis were direct observation in the field (*Swennen & Yu, 2005*) or microhistologic analysis of remnants in feces and/or gut contents (*James & Burney, 1997*; *Fox et al., 2007*). While these approaches have been proved useful in some cases, they are relatively labor-intensive and greatly skill-dependent (*Fox et al., 2007*; *Samelius & Alisauskas, 1999*; *Symondson, 2002*). Applications of other methods for analyzing gut contents or feces were also restricted due to inherent limitations, as reviewed by *Pompanon et al. (2012)*. Recently, metabarcoding methods, based on high-throughput sequencing, have provided new perspectives for diet analysis and biodiversity assessment (*Taberlet et al., 2007*; *Creer et al., 2010*). These methods provide higher taxonomic resolution and higher detectability with enormous sequence output from large-scale environmental samples, such as soil, water and feces (*Shokralla, Spall & Gibson, 2012*; *Bohmann et al., 2014*). Owing to these advantages, metabarcoding has been widely employed in the diet analysis of herbivores (*Taberlet et al., 2012*; *Ando et al., 2013*; *Hibert et al., 2013*), carnivores (*Deagle, Kirkwood & Jarman, 2009*; *Shehzad et al., 2012*) and omnivores (*De Barba et al., 2014*). But the pitfalls of metabarcoding should not be ignored when choosing suitable techniques for new studies. For instance, many researches have shown that it is difficult

to obtain quantitative data using metabarcoding (*Sun et al., 2015*). This drawback might result from both technical issues of this method and relevant biological features of samples (*Pompanon et al., 2012*).

One paramount prerequisite of metabarcoding methods is to select robust genetic markers and corresponding primers (*Zhan et al., 2014*; *Zhan & MacIsaac, 2015*). For diet studies of herbivores, at least eight chloroplast genes and two nuclear genes are used as potential markers for land plants (*Hollingsworth, Graham & Little, 2011*). Although mitochondrial cytochrome *c* oxidase I (COI) is extensively recommended as a standard barcode for animals, its relatively low rate of evolution in botanical genomes precludes it being an optimum for plants (*Wolfe, Li & Sharp, 1987*; *Fazekas et al., 2008*). The internal transcribed spacer (ITS) is excluded due to divergence discrepancies of individuals and low reproducibility (*Álvarez & Wendel, 2003*). A variety of combinations and comparisons have been performed for the eight candidate genes, however, none proved equally powerful for all cases (*Fazekas et al., 2008*). Consequently, it is more effective to choose barcodes for a circumscribed set of species occurring in a regional community (*Kress et al., 2009*). Another equally important aspect of metabarcoding applications is the construction of reference libraries which assist taxonomic assignment (*Rayé et al., 2011*; *Xu et al., 2015*). It is difficult to accurately interpret sequence reads without a reliable reference library (*Elliott & Jonathan Davies, 2014*).

Diet analysis is one of the central issues in waterbird research, both for deciphering waterfowl population dynamics and interpreting inter- or intra-specific interactions of cohabitating species (*Zhao et al., 2015*). For instance, more than 60% of bean goose *Anser fabalis* and almost 40% of greater white-fronted goose *Anser albifrons* populations along the East Asian–Australian Flyway Route winter at the Shengjin Lake National Nature Reserve (*Zhao et al., 2015*). Previous studies based on microhistologic observation illustrated that the dominant composition of their diets was monocotyledons, such as *Carex* spp. (*Zhao et al., 2012*), Poaceae (*Zhang et al., 2011*), and a relatively small proportion of non-monocots (referred to as dicotyledons in the study of '*Zhao, Cao & Fox, 2013*'). However, few food items could be identified to species-level, mainly owing to variable tissue structures within plants, similar morphology between relative species, and a high level of degradation after digestion (*Zhang et al., 2011*; *Zhao et al., 2012*; *Zhao, Cao & Fox, 2013*). Ambiguous identification has hindered understanding of waterbird population dynamics and potential to establish effective conservation plans for them.

In this study, we aimed to improve this situation using the metabarcoding method to analyze diets of these species (see flowchart in Fig. 1). By examining the efficiency of eight candidate genes (*rbc*L, *rpo*C1, *rpo*B, *mat*K, *trn*H-*psb*A, *trn*L (UAA), *atp*F-*atp*H, and *psb*K-*psb*I), we selected robust genes and corresponding primers for reference library construction and high-throughput sequencing. Subsequently, we used the metabarcoding method to investigate diet composition of these two species based on feces collected from Shengjin Lake. Finally, we discussed and compared results from microhistology and DNA metabarcoding using the same samples to assess the utility and efficiency of these two methods.

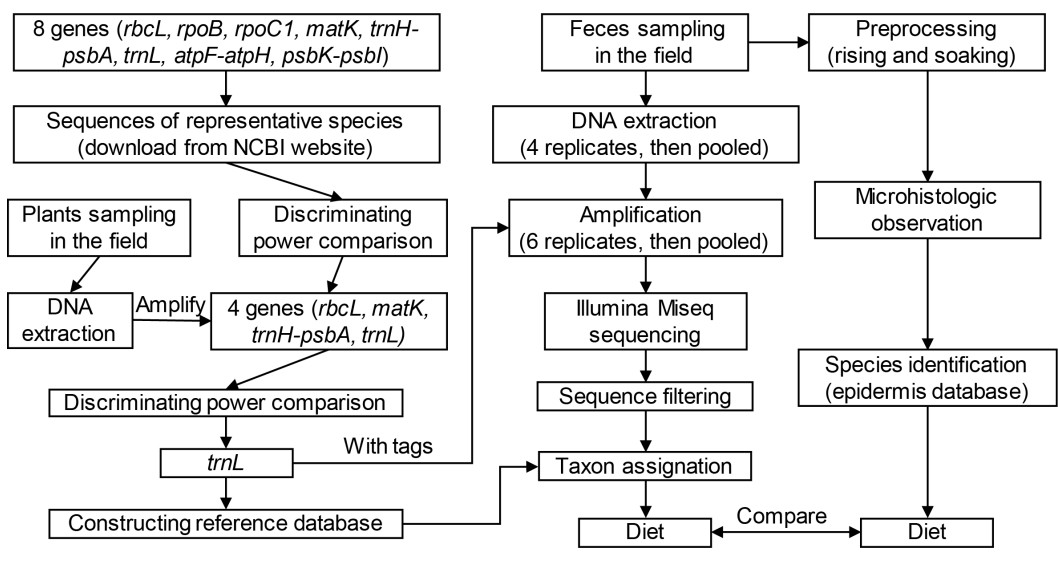

**Figure 1** Technical flowchart of this study.

# MATERIALS AND METHODS

## Ethics statement

Our research work did not involve capture or any direct manipulation or disturbances of animals. We collected samples of plants and feces for molecular analyses. We obtained access to the reserve under the permission of the Shengjin Lake National Nature Reserve Administration (Chizhou, Anhui, China), which is responsible for the management of the protected area and wildlife. We were forbidden to capture or disturb geese in the field.

## Study area

Shengjin Lake (116°55′–117°15′E, 30°15′–30°30′N) was established as a National Nature Reserve in 1997, aiming to protect waterbirds including geese, cranes and storks. The water level fluctuates greatly in this lake, with maximal water level of 17 m during summer (flood season) but only 10 m during winter (dry season). Due to this fluctuation, receding waters expose two large *Carex* spp. meadows and provide suitable habitats for waterbirds. This makes Shengjin Lake one of the most important wintering sites for migratory waterbirds (*Zhao et al., 2015*). Greater white-fronted goose and bean goose are the dominant herbivores wintering (from October to April) in this area, accounting for 40% and 60% of populations along the East Asian–Australian Flyway Route, respectively (*Zhao et al., 2015*).

## Field sampling

The most common plant species that these two geese may consume were collected in May 2014 and January 2015, especially species belonging to *Carex* and *Poaceae*. Fresh and intact leaves were carefully picked, tin-packaged in the field and stored at −80 °C in the laboratory before further treatment. Morphological identification was carried out with the assistance of two botanists (Profs. Zhenyu Li and Shuren Zhang from Institute of Botany, Chinese Academy of Sciences).

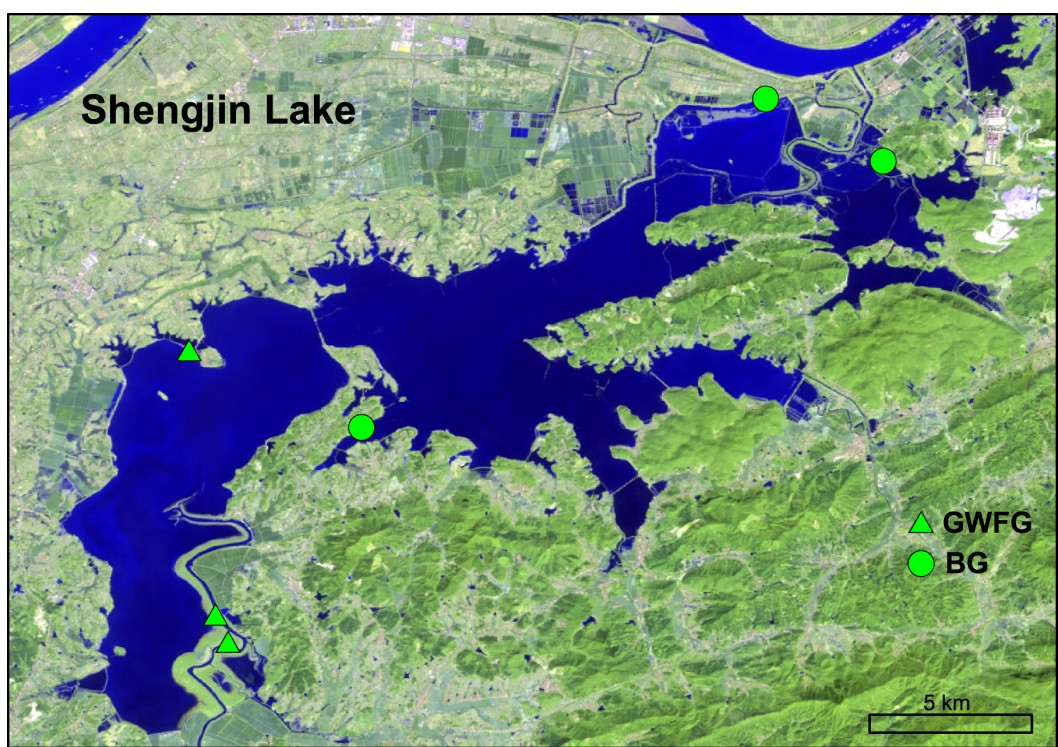

**Figure 2** **The location of our study area, Shengjin Lake National Nature Reserve and our sampling sites.** (Source: http://eros.usgs.gov/#).

All feces were collected at the reserve (Fig. 2) in January 2015. Based on previous studies and the latest waterbird survey, sites with large flocks of geese (i.e., more than 200 individuals) were chosen (*Zhang et al., 2011*). As soon as geese finished feeding and feces were defecated, fresh droppings were picked and stored on dry ice. Droppings of bean geese were generally thicker than those of smaller greater white-fronted goose, to the degree that these could be reliably distinguished in the field (*Zhao et al., 2015*). Disposal gloves were changed for each sample to avoid cross contamination. To avoid repeated sampling and to make sure samples were from different individuals, each sample was collected with a separation of more than 2 m. In total, 21 feces were collected, including 11 for greater white-fronted goose and 10 for bean goose. All samples were transported to laboratory on dry ice and then stored at −80 °C until further analysis.

**Selection of molecular markers and corresponding primers**

Here, we aimed to select gene markers with adequate discriminating power for our study. We included eight chloroplast genes— *rbc*L, *rpo*C1, *rpo*B, *mat*K, *trn*H-*psb*A, *trn*L (UAA), *atp*F-*atp*H, and *psb*K-*psb*I for estimation. Although Shengjin Lake included an array of plant species, we focused mainly on the most likely food resources (*Xu et al., 2008*; *Zhao et al., 2015*) that geese would consume for candidate gene tests. These covered 11 genera and the family *Poaceae* (Table S1). For tests of all candidate genes, we recovered sequences of representative species in the selected groups from GenBank (http://www.ncbi.nlm.nih.gov/nuccore). We calculated inter-specific divergence within

every genus or family based on the Kiruma 2-parameter model (K2P) using MEGA version 6 (*Tamura et al., 2013*). We also constructed molecular trees based on UPGMA using MEGA and characterized the resolution of species by calculating the percentage of species recovered as monophyletic based on phylogenetic trees (Rf). Secondly, primers selected out of eight candidate genes were used to amplify all specimens collected in Shengjin Lake and to check their amplification efficiency and universality. Thirdly, we calculated inter-specific divergence based on sequences that we obtained from last step. Generally, a robust barcode gene is obtained when the minimal inter-specific distance exceeds the maximal intra-specific distance (e.g., existence of barcoding gaps). Finally, to allow the recognition of sequences after high-throughput sequencing, both of the forward and reverse primers of the selected marker gene were tagged specifically for each sample with 8nt nucleotide codes at the 5′end (*Parameswaran et al., 2007*).

## DNA extraction, amplification and sequencing

Two hundred milligrams of leaf was used to extract the total DNA from each plant sample using a modified CTAB protocol (*Cota-Sanchez, Remarchuk & Ubayasena, 2006*). DNA extraction of feces was carried out using the same protocol with minor modification in incubation time (elongate to 12 h). Each fecal sample was crushed thoroughly and divided into four quarters. All quarters of DNA extracts were then pooled together. DNA extraction was carried out in a clean room used particularly for this study. For each batch of DNA extraction, negative controls (i.e., extraction without feces) were included to monitor possible contamination.

For plant DNA extracts, PCR amplifications were carried out in a volume of 25 μl with ∼100 ng total DNA as template, 1U of *Taq* Polymerase (Takara, Dalian, Liaoning Province, China), 1× PCR buffer, 2 mM of $Mg^{2+}$, 0.25 mM of dNTPs, 0.1 μM of forward primer and 0.1 μM of reverse primer. After 4 min at 94 °C, the PCR cycles were as follows: 35 cycles of 30 s at 94 °C, 30 s at 56 °C and 45 s at 72 °C, and the final extension was 10 min at 72 °C. We applied the same PCR conditions for all primers. All the successful PCR products were sequenced with Genewiz (Suzhou, Jiangsu Province, China).

For fecal DNA extracts, PCR mixtures (25 μl) were prepared in six replicates for each sample to reduce biased amplification. Each replicate was subjected to the same amplification procedure used for plant extracts. The six replicates of each sample were pooled and purified using the Sangon PCR product purification kit (Sangon Biotech, Shanghai, China). Quantification was carried out to ensure equilibrium of contribution of each sample using the NanoDrop ND-2000 UV-Vis Spectrophotometer (NanoDrop Technologies, Wilmington, Delaware, USA). High-throughput sequencing was performed using Illumina MiSeq platform following manufacturer's instructions by BGI (Shenzhen, Guangdong Province, China). Reads of high-throughput sequencing could be found at NCBI's Sequence Read Archive (Accession number: SRP070470).

## Data analysis for estimating diet composition

After high-throughput sequencing, pair-ended reads were merged with the fastq_mergepairs command using usearch (http://drive5.com/usearch, *Edgar, 2010*). Reads were then split

into independent files according to unique tags using the initial process of RDP pipeline (https://pyro.cme.msu.edu/init/form.spr). We removed sequences (i) that didn't perfectly match tags and primer sequences; (ii) that contained ambiguous nucleotide (N's). Tags and primers were then trimmed using the initial process of RDP pipeline. Further quality filtering using usearch discarded sequences with (i) quality score less than 30 (<Q30) and (ii) length shorter than 100 bp. Unique sequences were clustered to operational taxonomy units (OTUs) at the similarity threshold of 98% (*Edgar, 2013*). All OTUs were assigned to unique taxonomy with local blast 2.2.30+ (*Altschul et al., 1990*). We detected a plant within the reference library for each sequence with the threshold of length coverage >98%, identity >98% and $e$-value $< 1.0e^{-50}$. If a query sequence matched two or more taxa, it was assigned to a higher taxonomic level which included all taxa.

### Microhistology analysis

We used the method described by *Zhang et al. (2011)* to perform microhistologic examination of fecal samples. Each sample was first washed with pure water and filtered with a 25-μm filter. Subsequently, the suspension was examined under a light microscope at 10× magnification for quantification statistics and at 40× magnification for species identification. We compared photos of visible fragments with an epidermis database of plants from Shengjin Lake to identify food items (*Zhang et al., 2011*).

## RESULTS

### Selection of genes and corresponding primers and reference library construction

A total of 3,296 representative sequences were recovered from GenBank, ranging from 0 to 345 sequences per gene per taxon (Table S1). For *Eleocharis* and *Trapa*, only sequences of *rbc*L gene and *trn*L gene were retained, which makes it unfair to compare the efficiency and suitability of eight candidate genes. For the other ten taxa, *trn*L, *trn*H-*psb*A, *rbc*L and p*sb*K-*psb*I showed the largest inter-specific divergence in five, three, one, and one taxonomic groups, respectively. In addition, *trn*H-*psb*A, *atp*F-*atp*H, *trn*L and *psb*K-*psb*I showed the highest mean divergence in four, four, one and one taxonomic groups, respectively. However, given the small number of sequences and coverage of species, the suitability and efficiency of *atp*F-*atp*H and *psb*K-*psb*I seem to be less reliable than others. This comparison makes *trn*H-*psb*A, *trn*L and *rbc*L to be selected out of the eight candidate genes. As *mat*K used to be recommended as the standard barcode gene for *Carex* species (*Starr, Naczi & Chouinard, 2009*), which happened to be the dominant food for herbivorous geese in our study (*Zhao et al., 2015*), we included *mat*K as a supplement at last.

Primers for these four genes (Table 1) were used to amplify the plants that we collected in the field. In total, we collected 88 specimens in the field, belonging to 25 families, 53 genera and 70 species (Table 2). The selected primers for *trn*L and *rbc*L successfully amplified 100% and 91% of all species, respectively, while primers for *trn*H-*psb*A and *mat*K amplified only 71% and 43%, respectively. Therefore, we chose *trn*L and *rbc*L to test their discriminating power in our target plants.

**Table 1  Primers of candidate genes and reference library constructing.** Only the *c* and *h* were used for high-throughput sequencing in fusion primer mode (primer + tags). The unique tags were used to differentiate PCR products pooled together for highthroughput sequencing (*Parameswaran et al., 2007*).

| Gene | Primer | Sequence (5′-3′) |
|---|---|---|
| *mat*K | matK-XF[a] | TAATTTACGATCAATTCATTC |
| | matK-MALP[b] | ACAAGAAAGTCGAAGTAT |
| *rbc*L | rbcLa-F[c] | ATGTCACCACAAACAGAGACTAAAGC |
| | rbcLa-R[d] | GTAAAATCAAGTCCACCRCG |
| *trn*H-*psb*A | pasbA3_f[e] | CGCGCATGGTGGATTCACAATCC |
| | trnHf_05[f] | GTTATGCATGAACGTAATGCTC |
| *trn*L | *c*[g] | CGAAATCGGTAGACGCTACG |
| | *h*[h] | CCATTGAGTCTCTGCACCTATC |

**Notes.**
[a] referred to *Ford et al. (2009)*.
[b] referred to *Dunning & Savolainen (2010)*.
[c] referred to *Hasebe et al. (1994)*.
[d] referred to *Kress et al. (2009)*.
[e] referred to *Tate & Simpson (2003)*.
[f] referred to *Sang, Crawford & Stuessy (1997)*.
[g] referred to *Taberlet et al. (1991)*.
[h] referred to *Taberlet et al. (2007)*.

We calculated the inter-specific divergence within genera and families with at least two species to compare their discriminating power. Maximal, minimal and mean inter-specific distances were calculated for seven dominant genera and six dominant families (Table 3). Neither gene could differentiate species of *Vallisneria* (mean = 0.000 ± 0.000%) or *Artemisia* (mean = 0.000 ± 0.000%). But *trn*L showed a larger divergence range for the other six genera and five families. Hence, we chose *trn*L as the barcoding gene for reference library constructing and high-throughput sequencing for our study. The discriminating power of *trn*L was strong for most species (Table 4). However, some species could only be identified at genus-level or family-level with *trn*L. For instance, five species of *Potamogetonaceae* shared the same sequences and this made them to be identified at genus-level. Species could be identified easily to genus and family, except for three grasses (*Poaceae*) *Beckmannia syzigachne*, *Phalaris arundinacea*, and *Polypogon fugax* which shared identical sequences.

## Data processing for estimating diet composition

In total, 0.21 and 0.18 million reads were generated for greater white-fronted goose (GWFG) and bean goose (BG), respectively (Table 5). The number of recovered OTUs ranged from 8 to 123 for GWFG and BG samples. We used local BLAST to compare these sequences with the Shengjin Lake reference database. Finally, with DNA metabarocoding, 12 items were discovered in the feces of GWFG, including one at family-level, three at genus-level and eight at species-level (Table 6). Four items were discovered in the feces of BG, including one at genus-level and three at species-level. In total, this method identified 15 taxa in feces of these geese.

**Table 2  Plant species in the reference library.** We collected these samples from Shengjin Lake.

| Species | No. of samples | Species | No. of samples |
|---|---|---|---|
| Curculigo orchioides | 1 | Trapella sinensis | 2 |
| Artemisia capillaris | 1 | Plantago asiatica | 1 |
| Artemisia selengensis | 2 | Alopecurus aequalis | 2 |
| Aster subulatus | 1 | Beckmannia syzigachne | 1 |
| Bidens frondosa | 1 | Bromus japonicus | 1 |
| Erigeron annuus | 1 | Cynodon dactylon | 2 |
| Gnaphalium affine | 1 | Phalaris arundinacea | 1 |
| Hemistepta lyrata | 1 | Phragmites australis | 1 |
| Kalimeris incisa | 1 | Poa annua | 1 |
| Bothriospermum kusnezowii | 1 | Polypogon fugax | 1 |
| Lobelia chinensis | 1 | Roegneria kamoji | 2 |
| Sagina japonica | 1 | Zizania latifolia | 1 |
| Stellaria media | 2 | Polygonum lapathifolium | 4 |
| Calystegia hederacea | 1 | Polygonum orientale | 1 |
| Cardamine lyrata | 1 | Polygonum perfoliatum | 1 |
| Carex heterolepis | 3 | Polygonum persicaria | 1 |
| Carex capricornis | 1 | Rumex trisetiferus | 3 |
| Carex paxii | 1 | Potamogeton crispus | 1 |
| Carex remotiuscula | 1 | Potamogeton maackianus | 1 |
| Fimbristylis dichotoma | 1 | Potamogeton malaianus | 1 |
| Eleocharis migoana | 1 | Potamogeton natans | 1 |
| Scripus karuizawensis | 1 | Potamogeton pectinatus | 1 |
| Nymphoides peltatum | 1 | Clematis florida | 1 |
| Myriophyllum spicatum | 1 | Ranunculus chinensis | 2 |
| Hydrilla verticillta | 1 | Ranunculus sceleratus | 2 |
| Hydrocharis dubia | 1 | Potentilla freyniana | 2 |
| Vallisineria spiralis | 1 | Gratiola japonica | 1 |
| Vallisneria spinulosa | 1 | Mazus miquelii | 2 |
| Juncus effusus | 1 | Veronica undulata | 1 |
| Juncus gracillimus. | 1 | Trapa bispinosa | 1 |
| Leonurus japonicus | 1 | Trapa maximowiczii | 1 |
| Salvia plebeia | 1 | Trapa pseudoincisa | 1 |
| Glycine soja | 1 | Trapa quadrispinosa | 1 |
| Vicia sativa | 1 | Hydrocotyle sibthorpioides | 1 |
| Euryale ferox | 1 | Torilis japonica | 2 |

However, the sequence percentage of each food item varied greatly (Table 6). For GWFG, the majority of sequences (96.36%) were composed of only five items—*Poaceae* spp. (47.98%, except *Poa annua*), *Poa annua* (21.86%), *Carex heterolepis* (17.51%), *Carex* spp. (9.01%, except *Carex heterolepis*), and *Alopecurus aequalis* (3.21%). For BG, almost all the sequences belonged to *Carex heterolepis* (99.49%). Other items only occupied a relatively small proportion of sequences. In addition, the presence of each item per sample was also unequal (Table S2). For example in GWFG, *Carex heterolepis*, *Carex* spp., *Poa*

**Table 3** Inter-specific divergences within dominant genera and families of *rbc*L gene and *trn*L gene with Kiruma 2-Parameter model. Underscores indicate the most common food composition based on earlier microhistologic analysis (*Zhao et al., 2012*; *Zhao et al., 2015*).

| Inter-specific divergence | Taxa | *rbc*L | | | *trn*L | | |
|---|---|---|---|---|---|---|---|
| | | Maximal | Minimal | Mean | Maximal | Minimal | Mean |
| Within genera | Artemisia | 0.000 | 0.000 | 0.000 ± 0.000 | 0.000 | 0.000 | 0.000 ± 0.000 |
| | Carex | 0.013 | 0.000 | 0.008 ± 0.006 | 0.058 | 0.000 | 0.027 ± 0.021 |
| | Polygonum | 0.027 | 0.000 | 0.010 ± 0.006 | 0.076 | 0.000 | 0.033 ± 0.022 |
| | Potamogeton | 0.012 | 0.000 | 0.005 ± 0.0034 | 0.016 | 0.000 | 0.005 ± 0.005 |
| | Ranunculus | 0.031 | 0.000 | 0.020 ± 0.009 | 0.042 | 0.021 | 0.024 ± 0.022 |
| | Trapa | 0.000 | 0.000 | 0.000 ± 0.000 | 0.081 | 0.000 | 0.049 ± 0.030 |
| | Vallisneria | 0.000 | 0.000 | 0.000 ± 0.000 | 0.000 | 0.000 | 0.000 ± 0.000 |
| Within families | Cyperaceae | 0.043 | 0.000 | 0.018 ± 0.010 | 0.178 | 0.000 | 0.084 ± 0.046 |
| | Asteraceae | 0.120 | 0.000 | 0.049 ± 0.017 | 0.087 | 0.000 | 0.023 ± 0.018 |
| | Poaceae | 0.025 | 0.000 | 0.016 ± 0.0009 | 0.166 | 0.000 | 0.074 ± 0.039 |
| | Hydrocharitaceae | 0.122 | 0.000 | 0.078 ± 0.020 | 0.159 | 0.000 | 0.100 ± 0.054 |
| | Polygonaceae | 0.043 | 0.000 | 0.020 ± 0.009 | 0.129 | 0.000 | 0.031 ± 0.022 |
| | Ranunculaceae | 0.033 | 0.016 | 0.017 ± 0.015 | 0.045 | 0.000 | 0.018 ± 0.013 |

**Table 4** Number of species and unique sequences for families with more than one species in Shengjin Lake plant database.

| Family | No. of species | No. of sequences |
|---|---|---|
| Asteraceae | 8 | 7 |
| Caryophyllaceae | 2 | 2 |
| Cyperaceae | 7 | 5 |
| Fabaceae | 2 | 2 |
| Hydrocharitaceae | 4 | 3 |
| Lamiaceae | 2 | 2 |
| Poaceae | 10 | 8 |
| Polygonaceae | 5 | 5 |
| Potamogetonaceae | 5 | 1 |
| Ranunculaceae | 3 | 3 |
| Scrophulariaceae | 3 | 3 |
| Trapaceae | 4 | 3 |
| Umbelliferae | 2 | 2 |

*annua* and *Potentilla supina* were present in almost all the samples, while *Stellaria media*, *Asteraceae* sp. and *Lapsana apogonoldes* occurred in only about one third of samples.

When the microhistologic examination was performed using the same samples, eight items were found in the feces of greater white-fronted goose, including one at family-level, four at genus-level and three at species-level (Table 6). Dominant items were *Poaceae* spp. (45.68%), *Alopecurus Linn.* (30.93%) and *Carex heterolepis* (16.39%). Seven items were found in the feces of bean goose, including four at genus-level and three at species-level (Table 6). Dominant items were *Carex heterolepis* (62.85%), *Asteraceae* sp. (14.55%), and *Alopecurus Linn.* (13.18%).

**Table 5  Summary of the process and results of high-throughput sequencing analysis.**

| Sample | Pair-end sequences | Sequences for which primers and tags were identified and with length >100 bp | Unique sequences | OTUs | Food items |
|---|---|---|---|---|---|
| GWFG1 | 16303 | 8627 | 1288 | 78 | 8 |
| GWFG2 | 25482 | 13449 | 1091 | 102 | 8 |
| GWFG3 | 19063 | 10056 | 1277 | 48 | 10 |
| GWFG4 | 23856 | 12548 | 1419 | 114 | 8 |
| GWFG5 | 20955 | 11249 | 1720 | 123 | 9 |
| GWFG6 | 11677 | 7205 | 973 | 52 | 9 |
| GWFG7 | 13377 | 6782 | 1328 | 59 | 9 |
| GWFG8 | 7749 | 3959 | 774 | 89 | 9 |
| GWFG9 | 16833 | 8799 | 1436 | 90 | 6 |
| GWFG10 | 18474 | 9819 | 449 | 32 | 9 |
| GWFG11 | 19648 | 10458 | 617 | 31 | 6 |
| BG1 | 20225 | 10254 | 784 | 23 | 4 |
| BG2 | 14195 | 7161 | 564 | 16 | 2 |
| BG3 | 2229 | 1149 | 255 | 12 | 4 |
| BG4 | 517 | 268 | 77 | 8 | 3 |
| BG5 | 28152 | 14033 | 1000 | 15 | 3 |
| BG6 | 16723 | 8484 | 740 | 17 | 4 |
| BG7 | 30166 | 15403 | 974 | 15 | 4 |
| BG8 | 30928 | 15706 | 1028 | 15 | 3 |
| BG9 | 8382 | 4489 | 446 | 13 | 4 |
| BG10 | 10714 | 5526 | 537 | 13 | 4 |

**Notes.**
GWFG, Greater white-fronted goose; BG, Bean goose.

# DISCUSSION

## Marker selection and reference library constructing for diet analysis

With greatly reduced cost, extremely high throughput and information content, metabarcoding has revolutionized the exploration and quantification of dietary analysis with noninvasive samples containing degraded DNA (*Fonseca et al., 2010*; *Shokralla et al., 2014*). Despite enormous potential to boost data acquisition, successful application of this technology relies greatly on the power and efficiency of genetic markers and corresponding primers (*Bik et al., 2012*; *Zhan et al., 2014*). In order to select the most appropriate marker gene for our study, we compared the performance of eight commonly used chloroplast genes (*rbc*L, *rpo*B, *rpo*C1, *mat*K, *trn*L, *trn*H-*psb*A, *atp*F-*atp*H, and *psb*K-*psb*I) and their corresponding primers. Although a higher level of discriminating power was shown in several studies, *atp*F-*atp*H and *psb*K-*psb*I were not as commonly used as other barcoding genes (*Hollingsworth, Graham & Little, 2011*). As one of the most rapidly evolving coding genes of plastid genomes, *mat*K was considered as the closest plant analogue to the animal barcode *COI* (*Hilu & Liang, 1997*). However, *mat*K was difficult to amplify using
**Table 6  List of the lowest taxonomic food items in the diet of geese.**

| Food items | Level of identification | GWFG | | | BG | | |
|---|---|---|---|---|---|---|---|
| | | N reads | $F_s$ (%) | $F_m$ (%) | N reads | $F_s$ (%) | $F_m$ (%) |
| *Poaceae* spp. (except *Poa annua*) | Family | 51705 | 47.98 | 45.68 | 0 | 0.00 | 0.00 |
| *Poa annua* | Species | 23554 | 21.86 | 0.00 | 167 | 0.20 | 0.00 |
| *Carex heterolepis* | Species | 18867 | 17.51 | 16.39 | 81457 | 99.49 | 62.85 |
| *Carex* spp. (except *Carex heterolepis*) | Genus | 9706 | 9.01 | 2.31 | 191 | 0.23 | 3.49 |
| *Alopecurus aequalis* | Species | 3458 | 3.21 | 0.00 | 0 | 0.00 | 0.00 |
| *Potentilla chinensis* | Species | 184 | 0.17 | 1.18 | 65 | 0.08 | 2.06 |
| *Cynodon dactylon* | Species | 155 | 0.14 | 0.00 | 0 | 0.00 | 0.00 |
| *Polygonum* spp. | Genus | 56 | 0.05 | 0.00 | 0 | 0.00 | 0.00 |
| *Stellaria media* | Species | 26 | 0.02 | 0.00 | 0 | 0.00 | 0.00 |
| *Ranunculus chinensis* | Species | 14 | 0.02 | 0.00 | 0 | 0.00 | 0.00 |
| *Lapsana apogonoides* | Species | 11 | 0.02 | 0.00 | 0 | 0.00 | 0.00 |
| *Asteraceae* sp. | Genus | 16 | 0.01 | 2.33 | 0 | 0.00 | 14.55 |
| *Alopecurus* | Genus | 0 | 0.00 | 30.93 | 0 | 0.00 | 13.18 |
| *Carex thunbergii* | Species | 0 | 0.00 | 0.54 | 0 | 0.00 | 2.79 |
| *Fabaceae* sp. | Genus | 0 | 0.00 | 0.64 | 0 | 0.00 | 1.08 |

**Notes.**
GWFG, Greater white-fronted goose; BG, Bean goose; $F_s$, percentage of sequences in DNA metabarcoding; $F_m$, percentage of epidermis squares in microhistological analysis.

available primer sets, with only 43% of successful amplification in this study. In spite of the higher species discrimination success of *trn*H-*psb*A than *rbc*L + *mat*K in some groups, the presence of duplicated loci, microinversions and premature termination of reads by mononucleotide repeats lead to considerable proportion of low-quality sequences and over-estimation of genetic difference when using *trn*H-*psb*A (*Graham et al., 2000*; *Whitlock, Hale & Groff, 2010*). In contrast, the barcode region of *rbc*L is easy to amplify, sequence, and align in most plants and was recommended as the standard barcode for land plants (*Chase et al., 2007*). The relatively modest discriminating power (compared to *trn*L) precludes its application for our study aiming to recover high resolution of food items. Consequently, *trn*L was selected out of eight candidate markers, with 100% amplification success, more than 90% of high quality sequences, and relatively large inter-specific divergence.

One of the biggest obstacles in biodiversity assessment and dietary analysis is the lack of a comprehensive reference library, without which it is impossible to accurately interpret and assign sequences generated from high-throughput sequencing (*Valentini, Pompanon & Taberlet, 2009*; *Barco et al., 2015*). In this study, we constructed a local reference library by amplifying the most common species (70 morpho-species in total) during the wintering period with the *trn*L gene. Although not all of them could be identified at species-level with *trn*L due to relatively low inter-specific divergence, many species could be separated with distinctive sequences. Previous studies have recommended group-specific barcodes to differentiate closely related plants at the species level (*Li et al., 2015*). For instance, *mat*K has been proved to be more efficient for the discrimination of *Carex* spp. (*Starr, Naczi & Chouinard, 2009*). However, the primer set of *mat*K failed to amplify species of *Carex* spp.

in our study, suggesting the universality of selected primer pairs should be tested in each study (*Zhan et al., 2014*).

## Applications of metabarcoding for geese diet analysis

A variety of recent studies have demonstrated the great potential of metabarcoding for dietary analysis, mainly owing to the high throughput, high discriminating power, and the ability to process large-scale samples simultaneously (*Creer et al., 2010*; *Taberlet et al., 2012*; *Shehzad et al., 2012*). In this study, we applied this method to recover diets of herbivorous geese and provided standard protocols for dietary analysis of these two ecologically important waterbirds. Our results further proved the more objective, less experience-dependent and more time-efficient character of DNA metabarcoding. However, not all the species in the reference library could be identified at species-level, owing to low inter-specific divergence. We suggest that multiple group-specific markers to be incorporated in the future, as in *De Barba et al. (2014)*. Two species, *Carex thunbergii* and *Fabaceae* sp., were only discovered via microhistologic analysis rather than metabarcoding. This failure might reflect the biased fragment amplification of current technology, of which dominant templates could act as inhibitors of less dominant species (*Piñol et al., 2015*). However, three species of *Poaceae* were only discovered using metabarcoding. In total, more taxa and higher resolution were attained using metabarcoding. But microhistology still proved a powerful supplementary. Previous studies using metabarcoding usually detected dozens of food items, even as many as more than one hundred species. For instance, 18 taxa prey were identified for leopard cat (*Prionailurus bengalensis*) (*Shehzad et al., 2012*); 44 plant taxa were recovered in feces of red-headed wood pigeon (*Columba janthina nitens*) (*Ando et al., 2013*); while more than 100 taxa were found in diet studies of brown bear (*Ursus arctos*) (*De Barba et al., 2014*). The relatively narrow diet spectrum of herbivorous geese may lead to misunderstanding that this result of our study is merely an artefact due to small sampling effort. However, this result is credible since these two geese species only feed on *Carex* meadow, where the dominant vegetation is *Carex* spp., with other species such as *Poaceae* and dicots (*Zhao et al., 2015*). Even though other wetland plants exist, they usually composed only a small proportion of the geese diets.

Quantification of food composition is another key concern in dietary analysis. Although the relative percentage of sequences was not truly a quantitative estimate of diet, taxa of the majority sequences in this study were in accord with microhistologic observations, which was considered an efficient way to provide quantitative results (*Wang et al., 2013*). Discrepancies might come from the semi-quantitative nature of metabarcoding methods (*Sun et al., 2015*). This is likely derived from PCR amplification, which always entails biases caused by universal primer-template mismatches, annealing temperature or number of PCR cycles (*Zhan et al., 2014*; *Piñol et al., 2015*). Other methods such as shotgun sequencing or metagenomic sequencing could be incorporated in the future to give information on abundances of food items (*Srivathsan et al., 2015*).

## Implications for waterbird conservation and wetland management

For long-distance migratory waterbirds, such as the wild geese in this study, their abundance and distribution are greatly influenced by diet availability and habitat use

(*Wang et al., 2013*). For example, waterbirds may be restricted at (forced to leave) certain areas due to favoring (loss) of particular food (*Wang et al., 2013*), while the recovery of such food may contribute to return of bird populations (*Noordhuis, Van der Molen & Van den Berg, 2002*). Results of both metabarcoding and microhistologic analysis in this study revealed that *Carex* and *Poaceae* were dominant food components which is in accordance with previous studies. The increasing number of these two geese wintering at the Shengjin Lake may be attributed to the expansion of *Carex* meadow, which offers access to abundant food resources (*Zhao et al., 2015*). Considering the long-distance migratory character of these birds, it is important to maintain energy balances and good body conditions in wintering areas because this might further influence their departure dates and reproductive success after arriving at breeding areas (*Prop, Black & Shimmings, 2003*). Based on this, it is important for wetland managers to maintain the suitable habitats and food resources for sustainable conservation of waterbirds, which highlights the significance of diet information. Our study also indicated that overlap and dissimilarity existed between the diets of these two geese. Animals foraging in the same habitats may compete for limited food resources (*Madsen & Mortensen, 1987*). This discrepancy of food composition may arise from the avoidance of inter-specific competition (*Zhao et al., 2015*). However, with the increase of these two species in Shengjin Lake, further research is needed to investigate the mechanisms of food resource partitioning and spatial distribution.

Shengjin Lake is one of the most important wintering sites for tens of thousands of migratory watebirds, while annual life cycles of these birds depend on the whole migratory route, including breeding sites, stop-over sites and wintering sites (*Kear, 2005*). Thus, a molecular reference library covering all the potential food items along the whole migratory route will be useful both for understanding of wetland connections and waterbird conservation. In addition, the ability of DNA metabarcoding to process lots of samples simultaneously enables rapid analyses and makes this method helpful for waterbird studies.

## ACKNOWLEDGEMENTS

We are very grateful to the stuff of the Shengjin Lake National Nature Reserve for their excellent assistance during the field work. Great thanks to Zhujun Wang and An An for feces collection in the field. We thank Song Yang for collecting plants in the Shengjin Lake Reserve. We also thank Profs. Zhenyu Li and Shuren Zhang for plant identification. Special thanks to Drs. Meijuan Zhao, Xin Wang, Fanjuan Meng and Peihao Cong for preparing the epidermis database and guiding microhistologic analysis.

### Funding

This work was supported by the National Basic Research Program of China (973 Program, Grant No. 2012CB956104) to E.Z., the National Natural Science Foundation of China (Grant No. 31370416), State Key Laboratory of Urban and Regional Ecology, Chinese Academy of Sciences (No. SKLURE2013-1-05) to L.C., the National Natural Science

Foundation of China (Grant No. 31500315) to X.W., and 100-Talent Program of the Chinese Academy of Sciences to A.Z. The funders had no role in study design, data collection and analysis, decision to publish, or preparation of the manuscript.

## Grant Disclosures

The following grant information was disclosed by the authors:
National Basic Research Program of China: 2012CB956104.
National Natural Science Foundation of China: 31370416.
State Key Laboratory of Urban and Regional Ecology.
Chinese Academy of Sciences: SKLURE2013-1-05.
National Natural Science Foundation of China: 31500315.

## Competing Interests

The authors declare there are no competing interests.

## Author Contributions

- Yuzhan Yang conceived and designed the experiments, performed the experiments, analyzed the data, contributed reagents/materials/analysis tools, wrote the paper, prepared figures and/or tables, reviewed drafts of the paper.
- Aibin Zhan conceived and designed the experiments, analyzed the data, contributed reagents/materials/analysis tools, reviewed drafts of the paper.
- Lei Cao conceived and designed the experiments, contributed reagents/materials/analysis tools, reviewed drafts of the paper.
- Fanjuan Meng and Wenbin Xu performed the experiments, contributed reagents/materials/analysis tools, reviewed drafts of the paper.

## Animal Ethics

The following information was supplied relating to ethical approvals (i.e., approving body and any reference numbers):
   Our research work did not involve capture or any direct manipulation or disturbances of animals.

## Field Study Permissions

The following information was supplied relating to field study approvals (i.e., approving body and any reference numbers):
   We collected samples of plants and feces for molecular analyses. We received permission to access the reserve for sampling from the Shengjin Lake National Nature Reserve Administration (Chizhou, Anhui, China), which is responsible for the management of the protected area and wildlife. We were forbidden to capture or disturb geese in the field.

## DNA Deposition

The following information was supplied regarding the deposition of DNA sequences:
   NCBI: Sequence Read Archive (SRA) database
   Accession number: SRP070470.

## Data Availability

The raw data has been supplied as a Supplemental Dataset.

## Supplemental Information

Supplemental information for this article can be found online at http://dx.doi.org/10.7717/peerj.2345#supplemental-information.

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
