# Peer review of "Selection of a marker gene to construct a reference library for wetland plants, and the application of metabarcoding to analyze the diet of wintering herbivorous waterbirds"

_PeerJ, doi:10.7717/peerj.2345_

## Round 0.1 · original submission · Minor Revisions

Only minor issues have been identified by the two referees below and these should not take you too long to address. I look forward to receiving the revision.

·

Basic reporting

The paper is well referenced and conforms to the PeerJ standard. Figures are relevant and adequate. Although the authors' intended meaning is reasonably clear throughout I felt that the English could be greatly improved through numerous minor corrections in word usage and order (Because there are many I have made suggested changes on the attached version of the manuscript for the authors to consider). In several instances hyperbole detracted from the presentation - I recommend removal of excess adjectives. There is some inconsistency in spelling, such as feces versus faeces (choose either American or standard English spelling). The use of species attributions (authorship) is highly inconsistent and adds unnecessary clutter. Table S1 is critical and should be in the main manuscript but could be improved by converting this to a simple species list and replacing replicate listings with a column for numbers of plants sampled.
The raw molecular data is available through the NCBI Sequence Read Archive SRP070470.

Experimental design

As original primary research in the field of molecular ecology, this is within the ambit of PeerJ. The questions addressed are clear (can metabarcoding improve on traditional methods for dietary analyses of migratory waterbirds, what are the best genes to use and what plant food resources are required for two species of geese at an important wintering site in central eastern China). Description of the analysis pipeline needs slight revision, specifying details (as noted in the manuscript). The % discrimination within groups (Rf) is a poor measure of resolvability because it depends on species sampling an phylogenetic context in each taxon - these qualifications on this measure should be stated. The arguments given for the choice of gene are inconsistent - in the end the choice is based largely on convenience (reasonably so). This could be made more explicit. Apart from this the experimental design is appropriate to the questions.

Validity of the findings

The data are robust and appropriately stated. I thought more could have been made of the striking difference in diet of the two species, and also, in cases where it was not possible to identify plants to species these might have been restricted to a subgeneric species cluster rather than a genus or family (e.g. Carex spp. excludes the most important species considered, Carex heterolepis).

Additional comments

While I do not agree with you that diet analysis is the most important aspect of understanding migratory bird conservation and population dynamics, I do see that it is an important aspect. All the best with your research and I hope to read subsequent papers following up on quantitative metabarcoding and on more extensive species analyses (waders with invertebrate diets?) and suggestions on how this can be applied to habitat augmentation or rehabilitation.

Reviewer 2 ·

Basic reporting

This study mainly deals with the comparison of the efficiency of eight potential metabarcoding markers for geese diet analysis. However, this aspect neither appeared in the title nor in the abstract. I suggest to include it both in the title and in the abstract.
Furthermore, the diet results are preliminary, and only validate the approach.

Experimental design

The experimental design has been done to validate the methods, not to really assess the diet of the two geese species.

Validity of the findings

Nothing wrong in the findings, but less emphasize must be given to the diet results according to the relatively low sample size.

Additional comments

Minor comments:
Line 21: the c/h trnL primers used here amplify a longer fragment than the P6 loop. The P6 loop is amplified with g/h primers (see Taberlet et al. 2007).
Table 1: the exact reference for the trnL c primer is "Taberlet P, Gielly L, Pautou G, Bouvet J (1991) Universal primers for amplification of three non-coding regions of chloroplast DNA. Plant Molecular Biology, 17, 1105–1109."

---

## Round 0.2 · Minor Revisions

I have made some very minor changes for the final submission - see attached file.

---

## Round 0.3 · accepted · Accept

Well done on a good MS. This is a nice contribution to the growing use of metabarcoding to determine diet.